# Telemedicine and its acceptance by patients with type 2 diabetes mellitus at a single care center during the COVID-19 emergency: A cross-sectional observational study

**Giovanni Sartore[1], Rosaria Caprino[1]\*, Eugenio Ragazzi[2], Annunziata Lapolla[1]**

**1** Department of Medicine–DIMED, University of Padua, Padua, Italy, **2** Department of Pharmaceutical and Pharmacological Sciences–DSF, University of Padua, Padua, Italy

\* dott.ssarosaria.caprino@gmail.com

**Data Availability Statement:** Data are available from Zenodo (DOI: 10.5281/zenodo.7010557).

## Abstract

### Introduction

When Italy was placed under lockdown to contain the COVID-19 pandemic from 9 March to 18 May 2020, alternative approaches to delivering care—such as telemedicine—were promoted for patients with chronic diseases like diabetes mellitus (DM). The aim of this study was to analyze patients' perception of, and satisfaction with the telehealth services offered during the COVID-19 emergency at an outpatient diabetes care unit in Italy.

### Methods

A cross-sectional survey was conducted on 250 patients with type 2 diabetes mellitus who regularly attended our diabetes care unit. Data were collected by means of telephone interviews, asking patients how they perceived the telehealth services, and their satisfaction with the televisit and computer-based care. A standardized questionnaire was administered: there were questions answered using a five-point Likert scale, and one open-ended question. Patients' demographic, anthropometric and biological data were collected from their medical records. Correlations between patients' characteristics, their perception of telemedicine, and their satisfaction with the televisit were examined. Spearman's rank-order correlation coefficient ρ (rho) and Kendall's rank correlation coefficient τ (tau) were used as nonparametric measures of the strength of the association between the scores obtained for the two ordinal variables, Perception and Satisfaction, and between other clinical parameters. Principal component analysis (PCA) was also used to assess overall links between the variables.

### Results

Almost half of the interviewees expressed a strongly positive perception of the medical services received, and more than 60% were very satisfied with the telehealth service provided during the COVID-19 emergency. There was a strong correlation between patients' perception and satisfaction ratings (*p*<0.0001). Duration of disease showed a significant positive

**Funding:** The authors received no specific funding for this work.

**Competing interests:** The authors have declared that no competing interests exist.

correlation with patients' satisfaction with their medical care. By means of PCA, it was found that BMI correlated inversely with both perception and satisfaction. Following a qualitative analysis of patients' answers to the open-ended question, contact with their specialist was important to them: it was reassuring and a source of scientifically correct information about their disease and the association between COVID-19 and diabetes.

## Conclusions

Based on our telephone interviews, patients appreciated the telehealth approach and were satisfied with it, regardless of the characteristics of their disease. Telemedicine proved essential to avoid interrupting the continuity of care, and therefore had not only clinical, but also psycho-social repercussions.

## Introduction

Diabetes mellitus (DM) has been defined as the second most common comorbidity in cases of COVID-19 [1], and research showed that diabetes negatively affects the prognosis in patients infected with COVID-19 [2,3]. Well-controlled blood glucose levels, maintaining glycemic variability within the range of 3.9 to 10.0 mmol/L, is associated with a significant reduction in the composite adverse outcomes and death [3]. On the other hand, poor glycemic control in patients with COVID-19 and pre-existing type 2 DM (T2DM) are associated with a worse outcome, a greater need for medical interventions, multi-organ injuries, and a higher mortality rate [4].

The COVID-19 pandemic did not only have an impact on people's physical health, it also prompted psychological repercussions–given its huge coverage by the mass media. Every country's citizens were experiencing anxiety, depression and stress–sometimes to such a degree that the psychological impact of lockdown was found to include post-traumatic stress, confusion and anger [5,6]. The strict emergency measures imposed to contain the virus suddenly upset people's normal lifestyles, with consequences on their psychological wellbeing and emotional states.

With the support of their health team, patients with diabetes have to take some responsibility for their own health and self-manage their medication. They can experience depression, anxiety, loneliness, worries and fears, and are liable to eating disorders, diabetes-related distress, and other issues associated with managing a chronic condition. Individuals with T2DM are up to twice as likely to have such psychosocial problems as members of the general population [7,8].

Joensen *et al.* [9] found that patients with diabetes who reported moderate-to-high levels of distress were more frequently concerned about "being overly affected by COVID-19 diabetes", or the fact that "people with diabetes are characterized as a risk group". They worried about "not being able to manage [their] diabetes if infected with COVID-19", about a "reduced quality of diabetes care" and "insufficient access to HCP (healthcare providers) if needed". Those who felt isolated and starved of company were also more likely to worry. Patients who missed having someone to talk to about their diabetes and felt lonely were also more likely to be worried. Another study [10], which involved the use of telemedicine, investigated the impact of living alone on glycemia in the context of anxiety associated with the risk of becoming infected with COVID-19. Positive correlations were found for fasting glycemia, duration of diabetes,

and postprandial glycemia in patients with high anxiety scores. Patients who were lonely emerged as a psychologically vulnerable population with poor glycemic outcomes. This study emphasized the importance of considering the influence of the COVID-19 pandemic and the overlap between glycemic and psychological aspects in a patient's management. Patients with diabetes have specific concerns regarding COVID-19 and their chronic disease, and this negatively affects their psychosocial health. These concerns should be addressed by providing people with diabetes answers to their specific questions and needs, and frequent updates on what we know about COVID-19 and diabetes. For diabetics, having access to emotional support [11] may be even more important during a pandemic, and telemedicine can be part of the solution [12,13]: it can reduce the anxiety of patients with T2DM, and thereby improve their glycemic control. Telehealth has an important role too, primarily in limiting the risk of exposure to the virus for patients and doctors, but also as a method for delivering several healthcare services.

Barr and Podolsky [14] wrote that epidemics, like wars, have the potential to catalyze and reconfigure developments in medicine and public health. During the COVID-19 pandemic, Webster [15] recorded a ten-fold increase in the use of telemedicine within a matter of weeks. For healthcare providers, this means developing skills in building virtual relationships, empathy, diagnostics, and providing advice to ensure a good-quality service, and planning efficient and effective interventions that benefit their patients. Care delivery modalities like telemedicine had already been promoted for the management of chronic diseases, especially DM, and they had been favorably received by patients [16]. Now the COVID-19 pandemic has made research on patient satisfaction in this setting much more important [17]. Few data are available as yet regarding this topic in patients with T2DM, particularly under lockdown conditions.

Italy was first placed under lockdown to contain the COVID-19 pandemic from 9 March to 18 May 2020. The COVID-19 emergency represented a moment of profound reflection for many healthcare sectors, and diabetologists had to focus on how best to manage patients who could no longer attend diabetes care centers. The question was whether telemedicine could serve as an integrated care-providing solution that would be acceptable to T2DM patients and enable the diabetes-related problems they experienced to be managed.

## Materials and methods

### Study design and population

A cross-sectional survey was conducted to examine patients' perception of, and satisfaction with the telemedicine service made available for telehealth consultations with the Public Diabetes Clinic of the ULSS 6 District Health Unit in Padua (north-east Italy) during a period of COVID-19 lockdown. The study was conducted in accordance with the Standards of Good Clinical Practice, as established by the Declaration of Helsinki. The study protocol was approved by the Ethics Committee for Clinical Trials in the Province of Padova, reference study No. 4964/U6/20. Verbal informed consent was given by all participants prior to their inclusion. All data were kept anonymized to protect participants' identity.

The study population consisted of adult patients (over 18 years old) with a history of T2DM, who had a telemedicine consultation (total number of televisits = 2463) with a diabetologist during the Italian COVID-19 lockdown from March to May 2020. Telemedicine consultation consisted in a telephone call that replaced the check-up visit, where the reports sent by e-mail were analyzed and a therapy confirmed or set.

Sample size was determined based on previous cross-sectional studies on telemedicine, reporting a high degree of satisfaction [18,19]. With a provisional estimate of 80% of satisfied/

very satisfied patients, a precision value of 0.05 and a Z-value of 1.96 (at 5% type I error), the formula [20] yielded a sample size of 246 subjects, that was rounded to 250.

Patients with the above characteristics were randomly selected and contacted, until reaching the scheduled sample size of 250. The number of patients selected was evenly distributed among the various diabetologists on the team to ensure that satisfaction with a given doctor was not a confounding factor.

The study was conducted between 21 July and 14 August 2020. Data were collected by means of a questionnaire during a telephone call made by a single operator, who took a neutral stance with the respondent. The operator stressed that the questionnaire was anonymous and explained that the purpose of the interview was for scientific research, not to assess the respondent's doctor. Three questions were used to assess patients' satisfaction: a Client Satisfaction Questionnaire (CSQ-3) was adapted for the purpose of discussing a telemedicine service, and also to promote the interviewees' adherence, and ensure that they understood the questions. The CSQ has shown a good reliability and validity, and it has been applied to a variety of hospital services [21–24]. Perception and satisfaction in the CSQ-3 was assessed using a five-point Likert scale [25].

Patients were asked about how they perceived the service: their attitude to it; how their expectations compared with the services received [26], and how they felt about telemedicine being used to manage DM. They were also asked about their satisfaction with the televisit, in terms of their emotions and individual experience of the healthcare provided during the telephone consultation. Each telephone interview lasted 15–20 minutes and was conducted in a room reserved at the outpatient clinic. Each variable considered in the questionnaire was scored from 1 to 5, where 1 meant a poor perception or satisfaction, and 5 meant a very strong perception or satisfaction. An open-ended question was then presented, asking respondents to provide feedback on the teleconsultation experience in their own words. Opinions on telemedicine were extracted and descriptively synthesized. They were divided into subgroups, then grouped into two broad categories, those in support and those against telemedicine.

Patients' demographic, biological and anthropometric data, including gender, age, duration of diabetes, glycated hemoglobin (HbA1c), body mass index (BMI), serum creatinine, and eGFR, were obtained from their medical records and used for the statistical analysis of the cohort.

## Data analysis

The data are presented as means ±SD, and as medians with interquartile ranges (IQR, calculated as the difference between 75th and 25th centiles). Spearman's rank-order correlation coefficient ρ (rho), and Kendall's rank correlation coefficient τ (tau) were used as nonparametric measures of the strength of the association between the scores obtained for the two ordinal variables, Perception and Satisfaction. Spearman's rank-order correlation coefficient was also calculated to compare the scores versus the other variables available, such as the patients' clinical and metabolic parameters. Statistical significance was assumed for $p < 0.05$.

The principal component analysis (PCA), a multivariate method of data evaluation, was applied to the previously mentioned data to obtain an overall picture of any links between the variables. The resulting biplot graph shows the score plot and the loading plot for the first two principal components. For the loading plot, the angles between the vectors indicate the degree of correlation with one another. For vectors forming a closed angle, the variables they represent are positively correlated. Conversely, for divergent vectors characterized by open angles nearing 180˚, the variables are negatively correlated; a 90˚ angle indicates no correlation between two variables.

## Results

### Patients' perceptions and levels of satisfaction, and their correlation with other variables

During the Italian lockdown in the spring of 2020, there were 2,463 televisits conducted for patients with T2DM. From among these cases, 250 patients were enrolled in the study and interviewed (Fig 1).

Table 1 shows these patients' main characteristics, and their perception and satisfaction scores.

Fig 2A shows how patients perceived their televisit. Nearly half the cohort (49.0%) awarded it a maximum score of 5, and for another 23.0% it scored 4 out of 5. In general, the higher the score from 1 to 5 (on a scale of increasing satisfaction from "not at all positive" to "very positive"), the higher the proportion of patients awarding it for their perception of the televisit.

Fig 2B shows that patient satisfaction was also generally high, when referring to the doctor's performance: 24.0% of the sample reported being satisfied (score 4) and 64.0% were very satisfied (score 5).

Fig 3 shows the strong correlation between the perception and satisfaction parameters ($p < 0.0001$). Spearman's rank-order correlation coefficient ρ (rho) was used as a nonparametric measure of the strength of the association between the two variables. Kendall's rank correlation coefficient τ (tau) was calculated to confirm the ordinal association between the variables. Both coefficients were statistically significant ($p < 0.0001$).

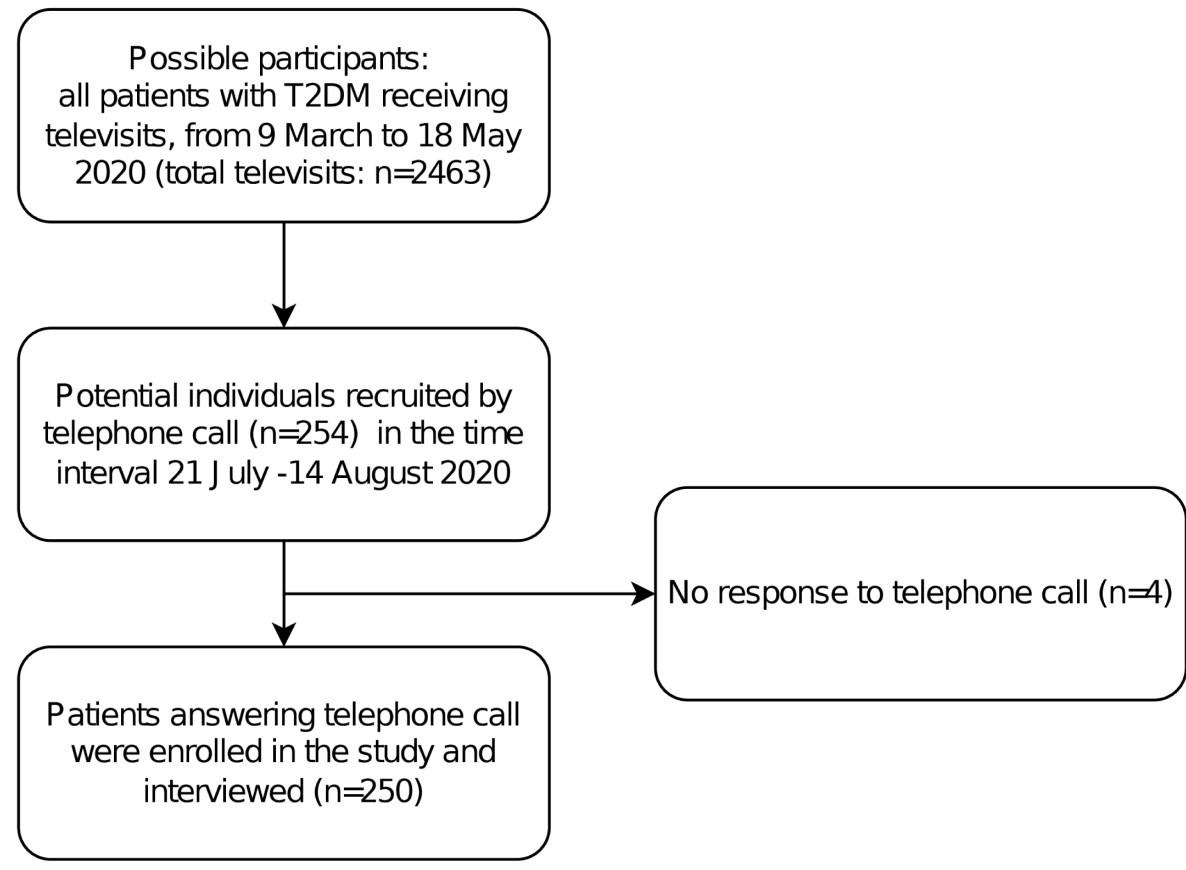

**Fig 1. STROBE diagram for various phases of the study.**

**Table 1. Characteristics of T2DM patients enrolled in the study (n = 250; male/female: 60%/40%).**

| Variable | mean ± SD | median (IQR) |
|---|---|---|
| Age, y | 70.1±11.0 | 71 (16) |
| Duration of diabetes, y | 14.2±9.6 | 13 (14) |
| HbA1c, % | 7.4±1.1 | 7.3 (1.3) |
| BMI, kg/m$^2$ | 29.5±4.8 | 29.1 (6.5) |
| Serum creatinine, mg/dL | 0.99±0.34 | 0.90 (0.40) |
| eGFR, mL/min ·1.73m$^2$ | 80.0±25.1 | 81.0 (37.0) |
| Perception of televisit (score 1 to 5) | 4.0±1.2 | 4 (2) |
| Satisfaction with televisit (score 1 to 5) | 4.4±0.9 | 5 (1) |

IQR: Interquartile range, calculated as the difference between the 75[th] and 25[th] centiles.

Table 2 shows the results regarding possible correlations between perception or satisfaction and age, HbA1c, BMI and duration of diabetes. The only significant correlation was between patients' satisfaction and the duration of their disease ($p$ = 0.0247).

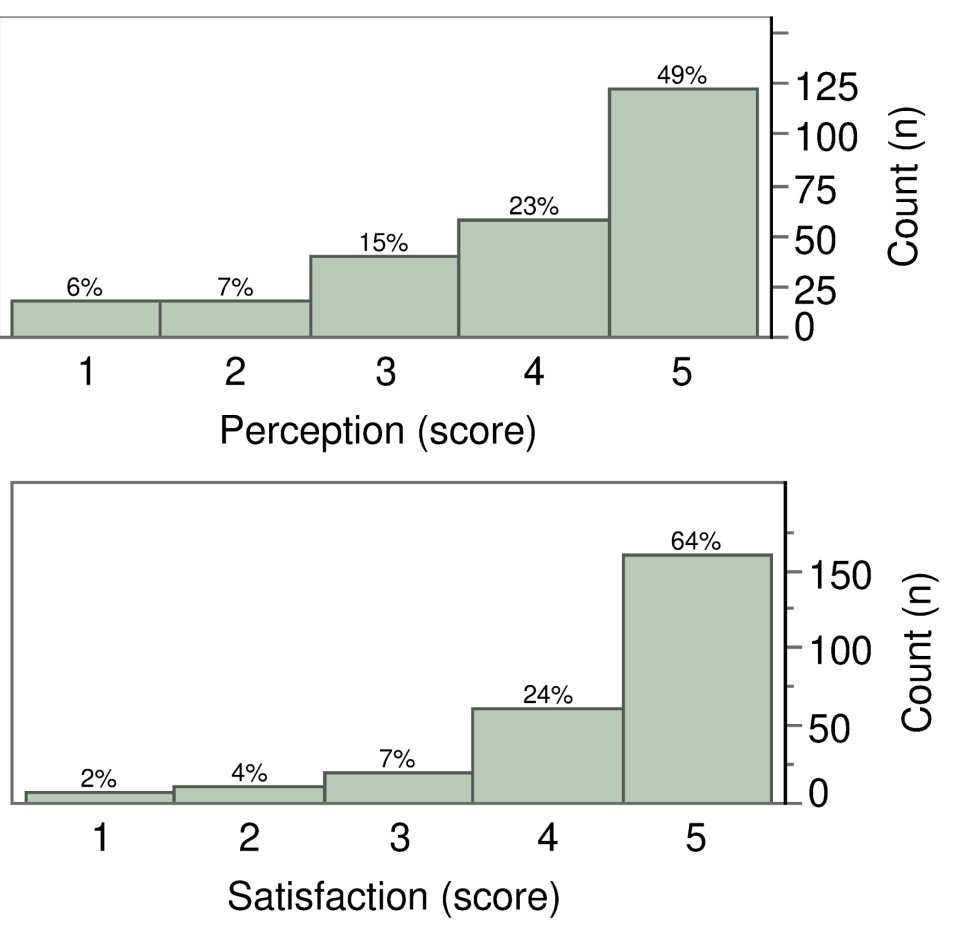

**Fig 2.** Distribution of the scores expressed by patients regarding their "Perception" of (A) and "Satisfaction" with (B) the telemedicine service. Scores were given on a five-point scale, from 1 (very low) to 5 (very high). The percentage of the sample is given above each bar.

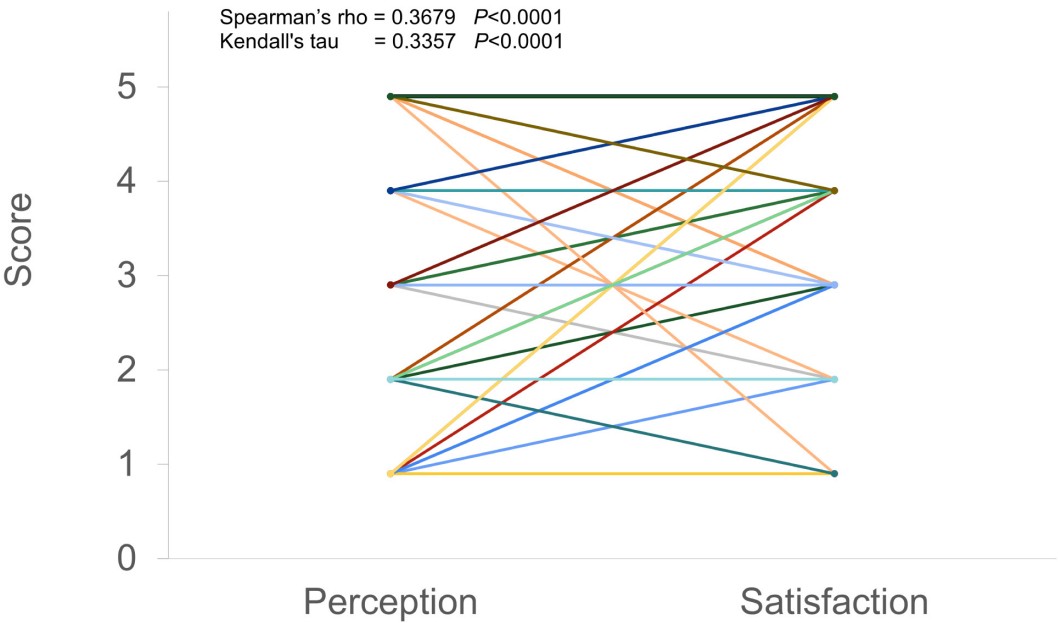

Spearman's rho = 0.3679   *P*<0.0001
Kendall's tau       = 0.3357   *P*<0.0001

**Fig 3. Correlation between "Perception" and "Satisfaction".** Spearman's rank-order correlation coefficient ρ (rho) was used as a nonparametric measure of the strength of the association between the two variables. Kendall's rank correlation coefficient τ (tau) was calculated to confirm the ordinal association between the variables. Both coefficients were statistically significant (*p*<0.0001).

A further analysis of data was obtained by means of PCA, a multivariate method that gives a general overview of underlying correlations that exist in a set of variables. When PCA was applied to the previously considered parameters, an inversely proportional correlation emerged between eGFR and HbA1c, and between eGFR and age (see the biplot graph in Fig 4). A partial correlation was found for both perception and satisfaction in relation to duration of disease, confirming the above-mentioned nonparametric finding. PCA also confirmed the

**Table 2. Correlation between scores (1 to 5) obtained for "Perception" and "Satisfaction" *versus* the variables considered.**

| 1ˢᵗ variable   *vs*   2ⁿᵈ variable | | Spearman's rho | *p* |
|---|---|---|---|
| Perception | Age, y | -0.0531 | 0.4035 |
| | Duration of diabetes, y | 0.0854 | 0.1781 |
| | HbA1c, % | 0.0325 | 0.6325 |
| | BMI, kg/m$^2$ | -0.0527 | 0.4357 |
| | Serum creatinine, mg/dL | -0.0402 | 0.5571 |
| | eGFR, mL/min ·1.73m$^2$ | 0.0417 | 0.5430 |
| Satisfaction | Age, y | -0.0196 | 0.7574 |
| | Duration of diabetes, y | 0.1420 | **0.0247*** |
| | HbA1c, % | 0.0462 | 0.4966 |
| | BMI, kg/m$^2$ | -0.0523 | 0.4394 |
| | Serum creatinine, mg/dL | 0.0577 | 0.3985 |
| | eGFR, mL/min ·1.73m$^2$ | -0.0565 | 0.4098 |

(*) Statistically significant value.

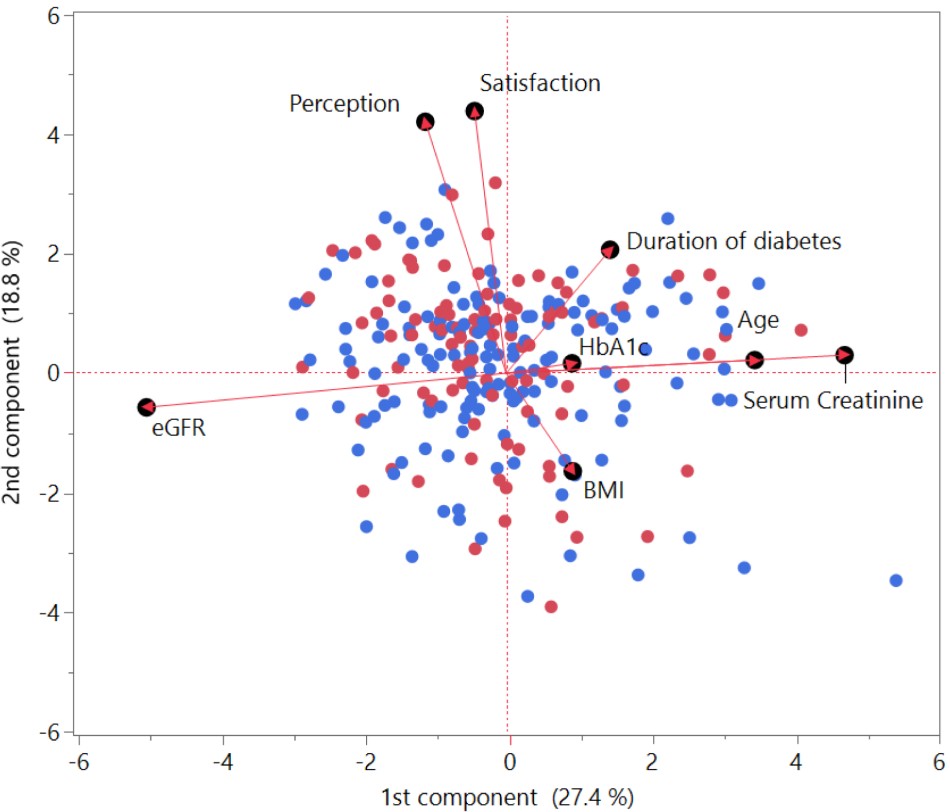

**Fig 4. Biplot of the first two components (on correlations) in the principal component analysis.** The Perception and Satisfaction variables were closely related, and the correlated moderately with the duration of diabetes, and inversely with BMI. According to this non-parametric analysis, there were no other correlations between the scores for Perception or Satisfaction and the other variables. Red dots indicate female patients, and blue dots indicate male patients; there were no distinctive clusters by sex.

close association between patients' perception of, and satisfaction with the televisit, which both appeared to be correlated inversely with their BMI.

## Patients' opinions

During the telephone interviews, patients were asked what they thought about their televisit experience. The points raised against telemedicine were a lack of direct face-to-face contact with the physician, and organizational problems (when booking another visit, for instance). The strengths patients identified were much more numerous. They were particularly grateful that the diabetes care service was not interrupted by the pandemic. They were relieved that they could still talk to their diabetologist, and access scientifically reliable information about the correlation between diabetes and COVID-19. Even patients unwilling to go to a hospital or clinics for fear of the infection were still able to have a doctor's appointment. Patients often described the televisit as being as good as a face-to-face visit. Other considerations concerned the benefits of saving time, not having to queue, and not having to look for a car parking space.

## Discussion

The use of telehealth services to support and promote long-distance clinical care, education, and health management plays a key role in disaster response [27]. In several ways, the

COVID-19 pandemic could be compared with a hurricane (particularly during lockdowns), and its impact could be particularly significant and persistent on healthcare delivery for patients with chronic diseases.

In our experience, using telemedicine to connect physicians and patients with T2DM via emails, phone calls, and text messages on mobile phones enabled patients to receive appropriate medical care. This approach was very positively perceived by our diabetic patients, whose satisfaction–regardless of their clinical features—confirmed recent findings concerning other chronic diseases [19,28,29].

An exception, highlighted thanks to the multivariate technique of PCA, was the inverse correlation emerging between patients' perception and satisfaction and their BMI. In other studies [30–32], obese people were reportedly less satisfied because they felt they had benefited less from their telemedical appointments than from face-to-face visits. This was probably because their therapy was closely linked to their lifestyle. Under lockdown conditions, the psychological stress of home confinement might also have triggered the onset or re-emergence of problematic eating patterns that, combined with a more limited physical exercise, would have resulted in unwanted weight gains [28]. Social distancing and lockdowns have influenced the general population's eating habits and physical activity levels too, but obese patients are those who have suffered the most [31]. A study by Athanasiadis *et al.* examined 208 patients who had previously undergone bariatric surgery, and they had all gained weight during the COVID-19 pandemic [32]. Apart from the decrease in physical activity, other risk factors for weight gain during lockdown were a lower consumption of healthy foods, loss of control when eating, and an increase in binge eating and snacking [32]. Social distancing also reportedly reduces the fear of confronting and of being judged negatively by others [33]. In fact, living with a larger number of people was found to reduce the risk of maladaptive eating due to emotional distress [34]. In short, our obese diabetic patients may have been less satisfied with their televisit because face-to-face visits normally brought benefits not only by providing drug therapies, but also by prompting changes in lifestyle, which is something that telemedicine could not do during lockdown. Being diabetic in times of COVID-19 raises the risk of hospitalization by three times, but for people who are diabetic and obese the risk is 4.5 times higher. Diabetes and obesity are also often associated with other cardiovascular risk factors, so patients therefore require a comprehensive approach [35] that was not available to them during the first lockdown. It may be that future improvements to the quality of care via telemedicine will enable healthcare providers to develop tailored interventions to better support these patients.

It is worth noting the close correlation that emerged between patients' satisfaction with their televisits and the duration of their diabetes. Previous studies had also found a significant correlation between diabetic patients' satisfaction with their treatment and the duration of their disease. It may be that patients with more experience, higher self-efficacy, and a greater knowledge of self-management strategies would benefit more from telemedicine services, and this could be taken into account in the management of diabetes in future. Duration of disease may also relate to patients' acceptance of their chronic condition, as they learn to cope with it better. Accepting a chronic disease takes time, and diabetic patients have to understand the importance of self-care and self-management [36]. Patients get to know their therapeutic and behavioral needs, which include proper nutrition and physical exercise, but they also need the diabetologist's periodic reinforcement. Diabetes self-management is a continuous learning process, which also benefits patients' psychosocial aspects and emotional well-being.

In general, our sample's satisfaction could be justified because their televisits enabled their disease to be managed, stimulating their adherence to therapies, routine blood glucose monitoring, healthy eating habits, and exercise. The influence of an effective doctor-patient relationship in promoting emotional well-being and treatment adherence is well established

[37,38], and the quality of the doctor-patient relationship has been shown to predict a range of health outcomes in people with diabetes [39]. A patient-centered relationship is associated with better diabetes self-care, greater adherence to treatment and psychosocial outcomes, and lower levels of stress because it nurtures patients' perceptions of their ability to self-manage their condition. An effective doctor-patient relationship relies on good communications from the doctor, and the patients' involvement in decision-making processes regarding their treatment, making sure they understand the benefits and importance of therapy [40]. Our open-ended questions clearly demonstrated the strong impact of the doctor-patient relationship when patients assessed their satisfaction with their televisit. Most of them seemed satisfied with the service (regardless of their glycemic control), and the empathic exchange they had with their doctor, even via remote means. This would suggest that telemedicine can have an intrinsic value, whatever the disease involved, that stems from the doctor-patient relationship, which concerns not only the clinical care aspect of a medical visit, but also the patient's bio-psycho-social sphere.

Positive patients' perception of telemedicine observed in this study was in agreement with previous ones carried out both in pre-COVID-19 and during COVID-19 era [41–46]. Post-pandemic tele-consultation lessons strengthen its effective role, in particular in underserved and remote areas [47], but its potential has not been completely explored, especially from patients' perspective [48]. On the one hand, there is a need of establishing telemedicine guidelines, training of consultants and advancement in technology [46], on the other hand of preventing the overuse in normal setting.

The present survey has some limitations. The size of the study sample is limited by its single-centered nature, as well as by considering T2DM patients and by the number of patients interviewed. Lack of information such as the educational status and current occupation would indicate how tech savvy the patients would have been to participate in tele-health consultations. Moreover, the study was conducted during severe restrictions period, which could be a partial confounding factor.

The present study confirms the importance of assessing patients' perceptions of telemedicine in order to lay the foundations for its future development, ensuring that recent lockdown experiences serve as an opportunity to improve our understanding of clinical and practical issues for use in a post-COVID-19 future.

## Acknowledgments

The authors thank the Staff of the outpatients' clinics at the Diabetes Clinic of the ULSS 6 District Health Unit in Padua for their cooperation, in particular Silvia Barbieri, Giuseppe Bax, Stefano Bassan, Lara Benetollo, Barbara Bonsembiante, Antonella Bortoletto, Federica Capovilla, Maria Grazia Dalfrà, Luisa Delorenzi, Lara Fanicchi, Alessandra Gallo, Debora Maggiolo, Michela Masin, Silvia Minardi, Beatrice Moro, Gloria Muraro, Isabella Negro, Manuela Nogara, Silvia Pastrolin, Roberta Penello, Francesco Piarulli, Susanna Picci, Mirella Pinton, Carlo Tintori, Barbara Toninello and Debora Vianello, and all the patients who agreed to take part in the investigation.

## Author Contributions

**Conceptualization:** Giovanni Sartore, Annunziata Lapolla.

**Data curation:** Rosaria Caprino.

**Formal analysis:** Eugenio Ragazzi.

**Methodology:** Eugenio Ragazzi.

**Supervision:** Giovanni Sartore.

**Validation:** Giovanni Sartore, Annunziata Lapolla.

**Writing – original draft:** Giovanni Sartore, Rosaria Caprino, Eugenio Ragazzi.

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
