## [Decision Letter · Decision Letter 0]

28 Jun 2022

PONE-D-22-14329Telemedicine and its acceptance by patients with type 2 diabetes mellitus at a single care center during the COVID-19 emergency: A prospective observational studyPLOS ONE

Dear Dr. Rosaria Caprino

Thank you for submitting your manuscript to PLOS ONE. After careful consideration, we feel that it has merit but does not fully meet PLOS ONE’s publication criteria as it currently stands. Therefore, we invite you to submit a revised version of the manuscript that addresses the points raised during the review process.

We look forward to receiving your revised manuscript.

Kind regards,

Pracheth Raghuveer, MD, DNB

Academic Editor

PLOS ONE

Journal Requirements:

Additional Editor Comments: STROBE diagram needs to be added. There are many grammatical errors that need correction. The discussion needs improvement. 

Reviewers' comments:

Reviewer's Responses to Questions

**Comments to the Author**

1. Is the manuscript technically sound, and do the data support the conclusions?

Reviewer #1: Yes

Reviewer #2: Partly

Reviewer #3: Partly

2. Has the statistical analysis been performed appropriately and rigorously? 

Reviewer #1: Yes

Reviewer #2: Yes

Reviewer #3: I Don't Know

3. Have the authors made all data underlying the findings in their manuscript fully available?

Reviewer #1: Yes

Reviewer #2: Yes

Reviewer #3: Yes

4. Is the manuscript presented in an intelligible fashion and written in standard English?

Reviewer #1: Yes

Reviewer #2: Yes

Reviewer #3: Yes

5. Review Comments to the Author

Reviewer #1: 1. sample size could have been increased for better inference.

2. few grammatical mistakes are there , please correct it.

3. please clarify , wether during telemedicine patient can see (Video interaction) their physician or not.

Reviewer #2: 1. This is a single cohort study without a control group during the Covid-19 pandemic when in person services were interrupted due to restrictions.

2. While methodology is sound the paper needs to be viewed in perspective.

3. This being a simple observational study it is not understood in what way “Standards of Good Clinical Practice and ICH Harmonized Tripartite Guidelines” were applicable in this study. These guidelines are for clinical trials.

4. Some care is better than no care. People around the world were numbed during the unprecedented severe restrictions in this period.

5. In such an environment even tele-consultation would have been welcome. The study findings need to be interpreted against this background.

6. Perhaps another study using same methods but in a post-pandemic setting would complement this study. Can be a recommendation in this paper.

7. Post pandemic tele-consultation lessons learned could be useful for delivering medical care in underserved and remote areas. However, the temptation to overuse it in normal settings should be checked.

8. May consider discussing some of these issues in the paper.

Reviewer #3: Dear author,

Thank you for undertaking this research to analyse the perceptions of patients on telehealth.

Overall the article is reasonably well presented. However I suggest that the following issues be addressed.

1. The setting needs to be well explained. The nature of the clinical practice (teaching or non teaching clinic), the characteristics of patients that are captured in this clinical practice whether it is a public or a private practice and the population that the clinic covers.

2. A STROBE diagram seems essential. Ideally it could start from the number of patients that attended the clinic during the study period number of type 2 diabetics, number who underwent tele health consultations and the number of patients who took part in the study. It is interesting that all 250 consecutive participants consented for the study and participated without any issues.

3. Sample size calculation has not been addressed. Starting from the hypothesis of this study, the reason as to how the number 250 was arrived at needs to be described clearly.

4. Regarding the study design, I disagree if this is a cohort study. There is no follow-up of patients. It seems to be a cross sectional study looking at the perspectives of type 2 diabetics who have attended tele-health consultation

5. While collecting baseline characteristics, a few other details could have been added. For example the sex distribution of the participants. Also the educational status, current occupation would indicate how tech savvy the patients would have been to participate in tele-health consultations.

6. The fact that this is a mixed method study with a qualitative component (the last open ended question of the questionnaire) the responses are ideally described as themes, sub themes and specific quotes. Also in the methodology section, there is paucity of information as to how this qualitative data was analysed.

7. The manuscript lacks a limitations section- even the best of studies will have its own limitations. It needs to be acknowledged.

8. In the discussion section, the generisability of the results need to be discussed. If similar results are expected in other parts of Italy or around the world.

9. Is there any specific reason why only type 2 diabetics were chosen and not any other types of diabetics?

10. The discussion seems to be more focused on BMI and the perception. But it is to be emphasised that there was a lack of statistical significance between the two, though there seemed to be a similar trend.

In summary, the above issues need to be addressed before considering this manuscript for publication.

Thank you for giving me the opportunity to review this manuscript.

6. PLOS authors have the option to publish the peer review history of their article (what does this mean?). If published, this will include your full peer review and any attached files.

Reviewer #1: **Yes: **Divendu Bhushan

Reviewer #2: **Yes: **Professor Amitav Banerjee, MD

Reviewer #3: No

---

## [Author Response · Author response to Decision Letter 0]

19 Aug 2022

We thank the Editor and the Reviewers for careful evaluation of our study and for all their valuable suggestions in order to improve the scientific merit of the manuscript. Detailed answers to all the criticisms are reported in the document.

---

## [Decision Letter · Decision Letter 1]

16 Nov 2022

PONE-D-22-14329R1Telemedicine and its acceptance by patients with type 2 diabetes mellitus at a single care center during the COVID-19 emergency: A prospective observational studyPLOS ONE

Dear Dr. Caprino,

Thank you for submitting your manuscript to PLOS ONE. After careful consideration, we feel that it has merit but does not fully meet PLOS ONE’s publication criteria as it currently stands. Therefore, we invite you to submit a revised version of the manuscript that addresses the points raised during the review process.

We look forward to receiving your revised manuscript.

Kind regards,

Pracheth Raghuveer, MD, DNB

Academic Editor

PLOS ONE

Journal Requirements:

Reviewers' comments:

Reviewer's Responses to Questions

**Comments to the Author**

1. If the authors have adequately addressed your comments raised in a previous round of review and you feel that this manuscript is now acceptable for publication, you may indicate that here to bypass the “Comments to the Author” section, enter your conflict of interest statement in the “Confidential to Editor” section, and submit your "Accept" recommendation.

Reviewer #1: All comments have been addressed

Reviewer #3: All comments have been addressed

2. Is the manuscript technically sound, and do the data support the conclusions?

Reviewer #1: Yes

Reviewer #3: Yes

3. Has the statistical analysis been performed appropriately and rigorously? 

Reviewer #1: I Don't Know

Reviewer #3: I Don't Know

4. Have the authors made all data underlying the findings in their manuscript fully available?

Reviewer #1: Yes

Reviewer #3: Yes

5. Is the manuscript presented in an intelligible fashion and written in standard English?

Reviewer #1: Yes

Reviewer #3: Yes

6. Review Comments to the Author

Reviewer #3: Dear authors,

Thank you for addressing the queries / suggestions. I suggest that the following be clarified in the manuscript.

1. Being a cross-sectional study, the words "prospective" and "consecutive" can be removed. They are usually used in the context of a prospective cohort study. From your methodology section, it appears that you have chosen all 254 patients who attended the practice within the given time period.

2. In the STROBE diagram, there have been 2463 patients between March and May 2020, however only 254 between July and August 2020. Is there a reason why the numbers dramatically reduced?

3. As raised by one of the other reviewers, whether Telehealth consultation refers to a video call/consultation or just an audio telephonic call with email communication needs to be clarified better.

7. PLOS authors have the option to publish the peer review history of their article (what does this mean?). If published, this will include your full peer review and any attached files.

---

## [Author Response · Author response to Decision Letter 1]

22 Nov 2022

Reviewer #3: Dear authors,

Thank you for addressing the queries / suggestions. I suggest that the following be clarified in the manuscript.

1. Being a cross-sectional study, the words "prospective" and "consecutive" can be removed. They are usually used in the context of a prospective cohort study. From your methodology section, it appears that you have chosen all 254 patients who attended the practice within the given time period.

We thank the Reviewer for suggestion. We deleted the words not appropriate for the study, and changed also STROBE diagram in order to make clearer the design of the survey.

2. In the STROBE diagram, there have been 2463 patients between March and May 2020, however only 254 between July and August 2020. Is there a reason why the numbers dramatically reduced?

The STROBE diagram was made clearer; the total number of televisits in the time intervals March - May 2020, corresponding to lockdown period, were 2463; among patients attending televisits, the survey recruited 250 patients who agreed to respond to the questionnaire. 

3. As raised by one of the other reviewers, whether Telehealth consultation refers to a video call/consultation or just an audio telephonic call with email communication needs to be clarified better.

Telehealth consultation consisted of a telephone call. The information was explained better in the text.

---

## [Decision Letter · Decision Letter 2]

2 Feb 2023

Telemedicine and its acceptance by patients with type 2 diabetes mellitus at a single care center during the COVID-19 emergency: A cross-sectional observational study

PONE-D-22-14329R2

Dear Dr. Caprino,

We’re pleased to inform you that your manuscript has been judged scientifically suitable for publication and will be formally accepted for publication once it meets all outstanding technical requirements.

Kind regards,

Pracheth Raghuveer, MD, DNB

Academic Editor

PLOS ONE

Additional Editor Comments (optional):

Reviewers' comments:

Reviewer's Responses to Questions

**Comments to the Author**

1. If the authors have adequately addressed your comments raised in a previous round of review and you feel that this manuscript is now acceptable for publication, you may indicate that here to bypass the “Comments to the Author” section, enter your conflict of interest statement in the “Confidential to Editor” section, and submit your "Accept" recommendation.

Reviewer #1: All comments have been addressed

Reviewer #3: All comments have been addressed

2. Is the manuscript technically sound, and do the data support the conclusions?

Reviewer #1: Partly

Reviewer #3: Yes

3. Has the statistical analysis been performed appropriately and rigorously? 

Reviewer #1: Yes

Reviewer #3: I Don't Know

4. Have the authors made all data underlying the findings in their manuscript fully available?

Reviewer #1: Yes

Reviewer #3: Yes

5. Is the manuscript presented in an intelligible fashion and written in standard English?

Reviewer #1: Yes

Reviewer #3: Yes

6. Review Comments to the Author

Reviewer #1: Author has responded to all points raised last time and did required modifications. This manuscript is technically sound and data given support the conclusion. Although author mentioned the difficulties , these can be better summarized as barrier to the tele medicine method of consultation. Statistical analysis appear good to me, but expert opinion can be taken.

Reviewer #3: Thank you for addressing my queries and suggestions. I am happy to accept the manuscript with the existing corrections.

7. PLOS authors have the option to publish the peer review history of their article (what does this mean?). If published, this will include your full peer review and any attached files.

Reviewer #1: **Yes: **This article showed benefits and barriers of telemedicine consultations. During COVID era, it was used as a great tool to serve people. For looking satisfaction and more details of barriers , qualitative studies can be done. Divendu Bhushan

Reviewer #3: **Yes: **Vignesh Kumar Chandiraseharan

---

## [Editor Report · Acceptance letter]

7 Feb 2023

PONE-D-22-14329R2 

Telemedicine and its acceptance by patients with type 2 diabetes mellitus at a single care center during the COVID-19 emergency: A cross-sectional observational study 

Dear Dr. Caprino:

I'm pleased to inform you that your manuscript has been deemed suitable for publication in PLOS ONE. Congratulations! Your manuscript is now with our production department. 

Kind regards, 

on behalf of

Dr. Pracheth Raghuveer 

Academic Editor

PLOS ONE